# Circadian Clock in Muscle Disease Etiology and Therapeutic Potential for Duchenne Muscular Dystrophy

**DOI:** 10.3390/ijms25094767

**Published:** 2024-04-27

**Authors:** Tali Kiperman, Ke Ma

**Affiliations:** Department of Diabetes Complications & Metabolism, Arthur Riggs Diabetes & Metabolism Research Institute, Beckman Research Institute of City of Hope, Duarte, CA 91010, USA; tkiperman@coh.org

**Keywords:** circadian clock, muscle regeneration, myogenesis, satellite cells, muscular dystrophy, small molecule, drug development

## Abstract

Circadian clock and clock-controlled output pathways exert temporal control in diverse aspects of skeletal muscle physiology, including the maintenance of muscle mass, structure, function, and metabolism. They have emerged as significant players in understanding muscle disease etiology and potential therapeutic avenues, particularly in Duchenne muscular dystrophy (DMD). This review examines the intricate interplay between circadian rhythms and muscle physiology, highlighting how disruptions of circadian regulation may contribute to muscle pathophysiology and the specific mechanisms linking circadian clock dysregulation with DMD. Moreover, we discuss recent advancements in chronobiological research that have shed light on the circadian control of muscle function and its relevance to DMD. Understanding clock output pathways involved in muscle mass and function offers novel insights into the pathogenesis of DMD and unveils promising avenues for therapeutic interventions. We further explore potential chronotherapeutic strategies targeting the circadian clock to ameliorate muscle degeneration which may inform drug development efforts for muscular dystrophy.

## 1. Introduction

The circadian clock that generates daily rhythms in physiology is driven by an interlocking molecular network of transcriptional feedback loops coupled with intricate translational and post-translational control. The hierarchical organization of the clock circuit in mammals consists of a central clock residing in the suprachiasmatic nuclei (SCN) of the hypothalamus and peripheral clocks presenting ubiquitously in distinct tissues [1,2]. Within skeletal muscle, this time-keeping machinery imparts rhythmic oscillations of key physiological processes in structure, function, growth, and metabolism [3]. A large body of evidence indicates the pervasive control of this timing mechanism in skeletal muscle biology. Recent discoveries have revealed the involvement of circadian clock dysregulation in muscle disease etiology, particularly muscular dystrophy. Muscular dystrophies are a group of disorders characterized by skeletal muscle degeneration and weakness. Duchenne muscular dystrophy (DMD) patients present with early onset muscle wasting, with progression to mobility limitations and respiratory failure. Due to the high incidence of 1:5000, early age onset, and premature death of DMD patients, there is an urgent need to identify potential therapeutic targets. Ongoing efforts in identifying novel modulators of the circadian clock loop enable the possibility for pharmacological intervention of clock regulators for muscle disease applications [4,5,6,7,8,9,10]. New developments in this field have revealed the intricate link between circadian clock dysregulation with muscular dystrophy etiology, suggesting the molecular clock circuit as a potential target for muscular dystrophy drug development [11,12]. This review will delve into recent progress on clock function in skeletal muscle biology, which is applicable to muscle diseases, and further discuss an emerging concept toward modulating clock function for muscular dystrophy therapy.

## 2. Circadian Clock Regulation of Skeletal Muscle Structure and Function

The circadian clock generates ~24 h oscillations of a diverse array of output pathways in behavior and physiology, in accordance with a multitude of external or intrinsic entrainment cues [2,13,14]. The daily cycles of the circadian clock are driven by interlocking transcriptional and translational molecular feedback loops, together with post-translational regulatory mechanisms. The positive transcriptional arm of the molecular clock circuit consists of the helix–loop–helix transcription factor circadian locomotor output cycles kaput (CLOCK) and brain and muscle arnt-like 1 (Bmal1). These transcription activators form a heterodimer, and through binding to canonical E-box motifs (CACGTG), turn on gene transcription of its target promoters [15]. Direct transcription targets of CLOCK/Bmal1 in the clock circuit consist of repressive components that complete the negative feedback loop, including the period (Per1, Per2, and Per3) and cryptochrome (Cry1 and Cry2) genes. Per1/2/3 and Cry1/2 proteins are transcription repressors that directly interact with and inhibit CLOCK/Bmal1-controlled transcription activation following nuclear translocation. This negative feedback mechanism consequently shuts down clock transcription, while subsequent proteasome-mediated degradation of Per and Cry proteins results in re-activation, leading to rhythmic clock oscillation [2,13] (Figure 1).

The skeletal muscle possesses a cell-autonomous functional circadian clock, with a significant 3.4% of the skeletal muscle transcriptome displaying diurnal oscillation [16,17]. Studies dissecting specific components of the circadian clock in skeletal muscle to date have collectively demonstrated its critical role in modulating skeletal muscle structure, function, muscle mass maintenance, and tissue remodeling processes [18,19,20,21,22,23]. The essential clock activator, Bmal1, is highly expressed in skeletal muscle. Distinct experimental models revealed its function in modulating myogenic differentiation required for skeletal muscle growth and regeneration [18,24,25]. CLOCK and Bmal1 display similar functions in orchestrating stem cell behaviors to promote skeletal muscle regenerative repair [20]. Notably, loss of CLOCK or Bmal1 function in muscle led to severely disrupted myofilament architecture and sarcomeric structure with impaired force generation [18]. CLOCK/Bmal1 complex was found to occupy E-boxes within the core enhancer of myogenic regulatory factor MyoD which drives its circadian oscillatory expression. Restoration of CLOCK/Bmal1 function in *Bmal1*-null mice, specifically in their brains, restored circadian behavior but not their impaired locomotor activity, while only restoration in muscle tissue normalized locomotor activity without behavioral rhythmicity [23]. Consistent with Bmal1 function in modulating skeletal muscle mass, *CLOCK* mutant mice displayed similarly affected muscle structure, as demonstrated by disorganized myofilaments, reduction in muscle force generation, and exercise tolerance [18]. As both *Bmal1*-null and *CLOCK* mutants displayed disorganizations of their myofilaments and sarcomeric structures [26], these structural deficits may have contributed to the attenuated maximal force production observed in whole muscle or single-fiber preparations. It is also intriguing that, given the similar presentation of muscles’ structural and functional deficits in CLOCK and MyoD mutants, a CLOCK/Bmal1 link with myogenic development via modulation of MyoD may mediate these phenotypes [18]. Investigations of additional genetic manipulations of distinct molecular clock components in mice, including Rev-erbs, Pers, and Crys, corroborated the notion that the entire circadian clock circuit contributes to the temporal orchestration of processes required for mature myocyte development [27,28,29,30]. Global ablation of Cry2 enhanced Pax7 expression and satellite cell proliferation, leading to augmented regenerative myogenesis [27]. On the other hand, inhibition of Rev-erbα/β by an antagonist SR8278 stimulated myogenic repair after injury [29].

Accumulating studies to date have uncovered the significant involvement of the circadian clock in driving time-of-day influence on distinct aspects of muscle function, particularly the role of exercise-induced response which entrains muscle clocks [21,31,32,33]. Muscle strength in humans, as measured by maximal isometric strength, increases from morning to evening, and this diurnal pattern could be driven by the circadian clock [34,35,36,37]. Human studies indicate that disruption of the clock, frequently as a result of sleep disruption or deprivation, reduces maximal isometric strength [38,39,40]. Findings from genetic mutant models of clock components reinforce the notion that circadian control is required for skeletal muscle function [18,26]. *Bmal1* deficiency in adult skeletal muscle resulted in reduction in maximum specific tension, accompanied by calcification and decreased collagen content. Similarly, *CLOCK* mutant mice displayed a 1.8-fold reduced daily locomotor activity level relative to wild-type mice [26]. Additional clock mutant strains also displayed significantly attenuated muscle functions. Targeted deletion of *Per2* impaired locomotor performance, although without affecting muscle contractility or muscle morphology [41]. *Rev-erbα* ablation resulted in lower exercise capacity, due to inappropriately elevated autophagy that accelerated mitochondrial clearance, coupled with reduced mitochondrial biogenesis [42]. Despite our extensive knowledge regarding circadian clock function in muscle development and physiology, pharmacological interventions to date targeting clock modulation for skeletal muscle disease therapy remain scarce.

## 3. Circadian Clock in Muscle Stem Cell Biology

Satellite cells residing beneath the skeletal muscle sarcolemma are the major muscle stem cell (MuSC) source for regenerative repair upon muscle injury [43]. These satellite cells are characterized by their expression of the paired box protein Pax7 and their localization between sarcolemma myofibers and the surrounding extracellular matrix’s basal lamina [44,45]. Upon muscle injury, satellite cells become activated to exit their quiescent state, accompanied by Pax7 binding with target DNA motifs to initiate a myogenic differentiation cascade through the induction of myogenic regulatory factors, Myf5, MyoD, and myogenin [46,47,48]. In the orderly progression through myogenesis, myogenic progenitors differentiate to acquire mature myocyte features that ultimately become multi-nucleated myofibers, a process that is shared during embryonic development and adult tissue remodeling [19,48,49]. In disease conditions such as DMD, satellite cell functions display significant alternations in polarity and asymmetric division, leading to the accumulation of a stem cell pool while attenuating myogenic progenitor commitment [50,51,52]. Various key aspects of stem cell dysfunction, involving reduced mitochondrial capacity, reactive oxygen species stress, senescence, altered plasticity, and aberrant epigenetic regulation in DMD, may collectively contribute to the defect of myogenic progenitors at mounting effective muscle regeneration in DMD [52]. Dissection of MuSC regulation by the circadian clock provides yet another layer to our current understanding of satellite cell dysfunctions involved in muscular dystrophies (Figure 2).

Studies from independent groups have collectively established that the circadian clock controls key aspects of MuSC functions during muscle growth and repair [24,25,28,30,53,54]. The molecular clock network, as demonstrated by genetic manipulations of key clock modulators, is required for satellite cell proliferative expansion and coordination of MuSC properties in regenerative myogenesis [24,53,55]. Bmal1 is an integral component of the pro-myogenic response required for muscle repair [25]. Genetic loss of *Bmal1* in mice leads to reduced total muscle mass, which is due to, at least in part, the defective myogenic differentiation of *Bmal1*-deficient primary myoblasts [24,25]. Chatterjee et al. demonstrated that satellite cell number and proliferative activity were reduced in *Bmal1*-deficient mice after cardiotoxin injury, accompanied by markedly attenuated satellite cell marker gene Pax7 expression [24,25]. Moreover, myogenic differentiation of primary myoblasts isolated from these mice were impaired, together with the observation of a significantly lower growth rate and proliferation [18,56]. Thus, the *Bmal1*-null mice failed to mount a robust regenerative response with markedly impaired nascent myofiber formation upon injury induction [25]. In line with these findings, Zhu et al. reported a similarly defective muscle regeneration phenotype of mice with selective *Bmal1* ablation in satellite cells, although this study examined a distinct NAD+ metabolic mechanism in mediating *Bmal1*’s effect [57].

Opposing feedback arms of the clock circuit, clock activator *Bmal1* and its repressor *Rev-erbα*, exert antagonistic transcriptional controls of the Wnt pathway to modulate myogenic differentiation [24,25,53]. While *Bmal1* promotes satellite cell proliferative expansion in muscle regeneration [24,25], *Rev-erbα* displays an opposite effect in suppressing regenerative myogenesis [55]. Recent studies from independent groups further corroborated these original findings by uncovering additional modulatory clock components in myogenic regulation, including clock repressors Pers and Crys [28,30,57]. A novel clock regulator, the CLOCK-interacting protein circadian (*Cipc*) [58], was found to inhibit Bmal1/CLOCK transcriptional activity through direct interaction with CLOCK [58]. Interestingly, CIPC colocalizes with Pax7 in primary myoblasts and is expressed in differentiated myotubes. Deletion of the *Cipc* gene in satellite cells augmented Pax7- and MyoD-expressing myogenic progenitor populations., This resulted in enhanced centro-nucleated fibers with an increased cross-sectional area indicative of augmented muscle regeneration. In contrast, *Cipc* overexpression downregulated Pax7 expression, demonstrating that CIPC is a repressor of Pax7 and inhibits regenerative myogenesis [58]. Other circadian clock repressors also exert negative regulations of MuSC properties that inhibit activation and differentiation. Chatterjee et al. demonstrated the cell-autonomous inhibitory effects of *Rev-erbα* on the proliferation and differentiation of myogenic precursor cells, with inhibition of muscle regeneration likely mediated through modulation of proliferative pathways and the Wnt cascade [53]. The absence of Rev-erbα enhanced the proliferative growth of the myogenic precursor population by augmenting satellite cell proliferative expansion and myogenic progression following muscle injury [53]. In a comparable manner, the closely related Rev-erbβ suppresses myoblast differentiation through a direct Rev-erb response element (RORE)-mediated transcriptional repression mechanism [59]. Recently, opposite roles were reported for Cry1 and Cry2 regarding their myogenic modulatory activities. While Cry1 is an inhibitory factor of this process and its deficiency promotes myogenic differentiation, Cry2 functions to promote myogenesis with the lack of Cry2 inhibiting myogenic differentiation [30]. An interesting mechanism underlies Cry2 action in promoting myogenic progression. Interaction between Cry2 and Bclaf1 stabilizes cyclin D1- and Tmem176b (a transmembrane regulator of myogenic cell fusion)-encoding mRNAs, and this mechanism prevents premature cell cycle exit with resultant short myotubes due to inefficient fusion [30]. A more recent report proposed a comparable inhibitory role of Cry2, similar to Cry1, in myogenic differentiation [27]. Using skeletal muscle and satellite cell-specific *Cry2*-null mouse models, *Cry2* ablation in myogenic progenitors stimulates their proliferation and increases myosin heavy chain (MyHC) expression [27]. It remains to be explored whether the global deletion as compared to cell type-selective manipulation of *Cry2* accounted for the differing findings in these studies [27,30]. *Per1* and *Per2* have also been demonstrated to be required for circadian regulation of muscle regeneration. Genetic ablation of either *Per1* or *Per2* leads to shorter circadian periods with suppressed myoblast differentiation, due to the failure to promote transcriptional activation of insulin-like growth factor 2 (Igf2)’s enhancer and promoter, an autocrine factor stimulating myoblast proliferation and differentiation [28]. Mechanistically, Igf2 binds to a type I receptor that activates the p38α/β MAPK pathway [60], triggering downstream cell cycle exit and myogenic factor induction, leading to differentiation [61,62]. Katoku-Kikyo et al. demonstrated Per-dependent recruitment of RNA polymerase II with dynamic histone modifications at the Igf2 promoter and enhancer, suggesting initiation of MuSC differentiation with epigenetic priming [28]. Intriguingly, the functions of *Per1* and *Per2* in regulating this pathway are non-redundant. At nighttime, when *Per1* and *Per2* are highly expressed under normal light–dark conditions, muscle regeneration was found to be more robust than that of the daytime points [28,63].

Elucidating circadian clock function and clock-controlled pathways that enhance satellite cell exit from quiescence, proliferation and differentiation may inform strategies to target the circadian clock circuit to promote remodeling for disease interventions. Particularly, based on the recent progress toward understanding circadian modulation of MuSC behavior and regenerative myogenesis, targeting these clock-controlled mechanisms to induce satellite cell proliferation and differentiation may prove to accelerate the muscular injury repair process. Promoting regenerative capacity may ameliorate muscular dystrophy or related disease conditions with loss of muscle mass. Although an underexplored area at present, targeted interventions to modulate clock activity for disease applications may spur drug development efforts for muscular dystrophy, particularly as combination therapy with currently available first-line treatments [64,65,66].

## 4. Circadian Clock Dysregulation in Muscle Disease Etiology

### 4.1. Involvement of Clock Dysfunction in Muscle Diseases

Understanding how each circadian clock component, with its distinct clock-controlled output pathways, modulates satellite cell properties, including quiescence, renewal, proliferation, and differentiation, provides the mechanistic basis for targeted interventional strategies. Novel clock-targeting approaches, through potential pro-myogenic effects, could be ultimately applied to stimulate muscle regeneration in disease states. Recently, new studies have uncovered that dysregulation of the circadian clock may contribute to the development of various muscle diseases (Figure 3). Impaired clock function induced muscle atrophy, with loss of muscle size and strength and impaired exercise capacity and mobility [67]. There are disparate mechanisms underlying the modulatory function of the circadian clock in skeletal muscle that may contribute to the development of atrophy if disrupted. These could involve potential temporal regulations in protein synthesis [68,69,70], protein degradation pathways [71,72,73], and mitochondrial activity and energy production [74,75,76], besides the role of the clock in promoting myofiber development and maturation. The fractional protein synthesis rate in skeletal muscle demonstrated diurnal variations across a 24 h period in rats [68], and circadian clock modulation was postulated to underlie, at least in part, the variable anabolic responses observed in resistance exercise [69]. Following a bout of exercise, the timing of exercise and food intake influences muscle protein accumulation and growth [77]. In muscle atrophy induced by sciatic nerve ablation, denervation of muscle led to altered core clock gene expression, with nearly 70% of circadian genes having lost rhythmicity [78]. The Per2 gene was found to be upregulated in denervated muscle [78]. Interestingly, loss of *Per2* in mice resulted in increased protein synthesis accompanied by suppressed autophagy in skeletal muscle [79], suggesting that in contrast to Bmal1/CLOCK, this clock repressor could be a negative regulator of muscle mass. On the other hand, protein degradation pathways, particularly autophagy as indicated by the observed number of autophagosomes, exhibit circadian rhythmic regulation [71]. Rev-erbα within the negative arm of the clock feedback loop was reported to exert strong repression of protein degradation pathways including key steps involved in autophagic flux and atrogenes that mediate ubiquitin-mediated proteasome degradation [71,73].

In line with the findings of the significant impacts of clock-controlled mechanisms on muscle mass regulation, circadian clock dysfunction was reported in sarcopenic conditions. Sarcopenia is characterized by loss of muscle mass, function, and strength, with decreased myofiber size and number, associated with aging or immobility [80,81]. Sarcopenia predisposes the risk for cardiovascular disease, with devastating consequence for increased mortality rate [82,83]. Various studies analyzing the impact of circadian clock disruption by sleep deprivation and shift work uncovered its detrimental effect on the development of sarcopenia [40,84,85]. Investigation of night shift workers in Korea, with demonstrated disruption of circadian rhythm, revealed significantly elevated risk for developing sarcopenia as compared to day shift workers. The disease risk for irregularly scheduled workers was increased even higher [84]. Interestingly, altered sleep durations in women, either long or short durations, were associated with increased risk for sarcopenia [85]. In contrast to the effect of clock disruption on impairing muscle mass and inducing atrophy, maintaining proper circadian clock function was found to be able to mitigate the risk for sarcopenia by enhancing muscle growth and mass maintenance [81]. Interestingly, molecular clock dysregulations were found to be associated with congenital collagen VI (*COL6A1*) deficiency-related myopathy. COL6A1 is a major component of skeletal muscle’s extracellular matrix, required for correct muscle cell adhesion, stability, and regeneration [86]. Notably, *collagen VI*-null mice with a mutated *COL6A1* gene demonstrated differentially expressed clock genes correlated with heightened inflammatory cytokine activity [87]. Further corroborating this observation, *Bmal1*-deficeint mice showed profound deregulation of the *COL6A1* pathway and autophagy-related genes. These findings of circadian clock dysregulation in collagen VI myopathies suggest its potential contribution to the etiology or pathogenesis of at least this specific form of myopathy.

### 4.2. The Role of the Circadian Clock in Muscular Dystrophy Etiology and Pathogenesis

Muscular dystrophies are characterized by a loss of muscle mass and progressive muscular weakness [88,89]. Duchenne muscular dystrophy (DMD) is an X-linked recessive disorder caused by a truncated *dystrophin* (*DMD*) gene [90,91,92], frequently due to mutations of inherited or random translocations between chromosome 21 and the X chromosome. The resultant out-of-frame or nonsense mutations lead to prematurely truncated dystrophin protein translation, rendering it nonfunctional [91]. Under normal physiological conditions, dystrophin protein is localized within the inner surface of the sarcolemma by anchoring a group of proteins, namely the dystrophin-associated protein complex (DAPC) [93,94,95], with an intracellular actin cytoskeleton network. The DAPC complex is required for connecting the extracellular matrix and sarcolemma with the intracellular cytoskeleton which stabilizes muscle membrane. Dystrophin and DAPC are thus involved in the regulation of skeletal muscle’s contractile function, mediated by their critical role in maintaining sarcolemma stability [95]. In DMD, due to the loss of structural linkages between the extracellular matrix with the intracellular cytoskeleton anchoring the sarcolemma, myocytes become susceptible to membrane damage induced by mechanical stress, associated with normal ambulatory activity or exercise, leading to cell death with severe consequences [96].

To date, corticosteroids remain as the first-line clinical therapy treatments for muscular dystrophies. Largely experimental interventions, such as cell therapies, gene therapy for direct protein delivery, and genetic modulations, are under active development to meet an urgent clinical need [89,91]. Effective, scalable, small-molecule drug development for these conditions to prevent or delay the disease course is in dire need, particularly as an alternative combinatorial therapeutic option with steroids. Recent progress using distinct clock mutant animal models crossed with mdx mice, a pre-clinical dystrophin-deficient model for DMD, established the genetic basis for circadian clock interventions to mitigate the disease pathophysiology of muscular dystrophies [55,56]. Critical insights in understanding the mechanistic links between clock dysfunction in dystrophic disease and clock-controlled mechanisms applicable for disease therapy may provide the foundation for developing novel clock-targeted therapies.

Emerging reports implicate the involvement of circadian clock dysregulation in the etiology or progression of dystrophic diseases [55,56,97]. Mechanisms linking dystrophin, intracellular actin cytoskeleton dynamics, serum response factor (SRF)-mediated signaling, and the circadian clock have been studied (Figure 4). SRF is a key transcription factor that mediates serum-induced intracellular actin cytoskeleton dynamics for signal transduction to regulate cell growth, migration, and myocyte development [98]. Gerber et al. demonstrated that the actin cytoskeleton organization and SRF-mediated transcription activity display diurnal oscillations [99]. Interestingly, our group demonstrated that the SRF and myocardian-related transcription factor A (MRTF-A)-mediated transcription activation, in response to actin remodeling dynamics, transduce extracellular physical niche cues to entrain the circadian clock, mediated via clock gene regulation such as by Pers and Rev-erbα [100]. Discovery of this mechanism of clock entrainment by SRF/MRTF-A-mediated transcription thus connects clock oscillation with extracellular niche cues transduced by an intracellular cytoskeleton signaling cascade [99,100]. The loss of dystrophin in DMD, with the absence of its tethering to an intracellular actin skeleton network, could thus interrupt sarcolemma–cytoskeleton linkage to prevent SRF-mediated signal activation in myofiber which attenuates circadian clock function [99]. In mdx mice and DMD patients, gene expression analysis revealed that Bmal1, Cry1, and Cry2 were downregulated in the dystrophic muscles, together with altered myogenic pathway involving MyoD and myogenin [101]. Furthermore, the loss of *dystrophin*-mediated ECM–cytoskeleton linkage in the mdx mice was found to disrupt clock entrainment [102]. Notably, in dystrophin-deficient myotubes and dystrophic mouse models, RhoA-actin-SRF signal transduction, with known circadian entrainment function in the SCN, was altered [102]. Another study by Hardee et al. also observed attenuated oscillatory rhythms of core clock genes in mdx mice, along with altered circadian profiles of autophagy- and mitochondrial-related genes [103]. Collectively, these recent findings uncovered the dysregulation of circadian clock outputs in muscular dystrophy, although their functional impact on dystrophic disease pathophysiology remains to be further investigated.

Another recent study of a milder form of congenital muscular dystrophy due to dystrophin mutation, Becker muscular dystrophy (BMD) [104], pointed to a direct circadian clock connection with muscular dystrophy [105]. In patients with BMD, dystrophin level, as detected by immunohistochemistry, was significantly diminished in histological sections of the heart, as compared to the hypertrophic heart, from non-BMD subjects. Furthermore, dystrophin protein displayed robust diurnal variability in heart and skeletal muscles in mice, with high levels detected at day time points. Dystrophin expression in the heart was significantly attenuated in mice with cardiac-specific loss of *Bmal1* [105]. This intriguing finding raised a distinct possibility that circadian clock function could be directly involved in or contribute to the pathogenesis or severity of BMD through its regulation of the dystrophin gene. Given this intriguing new finding, a clock–dystrophin regulatory link requires additional independent experimental validations. Nonetheless, these studies uncovered a new direction for unraveling the mechanistic underpinnings of clock-related etiology involved in muscular dystrophy. Collectively, these findings of the clock–muscular dystrophy connection provide a strong rationale for exploring clock-augmenting interventions which may have ameliorative effects in reinforcing sarcolemma integrity, although the precise molecular mechanisms involved require comprehensive future investigations.

Most intriguingly, asynchronous regenerative foci were identified in dystrophic muscles as being associated with failure to repair damage and disease progression [106]. Steroid treatment, known to induce clock synchronization [107], was able to mitigate the dyssynchrony of injury sites, suggesting that potential temporal coordination of regenerative loci may facilitate dystrophic repair [108]. In addition, dystrophic models showed impaired mitochondrial content and function which may contribute to disease severity and progression [109], and mitochondrial energy metabolism is subjected to clock regulation [110,111]. Loss of clock regulators in mice caused them to display an altered mitochondrial phenotype in their skeletal muscle, demonstrating that circadian clock dysfunction contributes to attenuated mitochondrial metabolism and quality control [42,112,113,114]. It is thus conceivable that in dystrophic muscles, maintaining or reinforcing a functioning clock machinery may protect energetic homeostasis with potential therapeutic values [103,115].

Chronic cycles of myofiber degeneration and regeneration in muscular dystrophies result in the progressive replacement of muscle with fibrotic tissue, fat, and inflammatory infiltrates that severely restrict contractile function [116,117]. Muscle regeneration in a dystrophic background is affected by attenuated satellite cell function [48]. Consistent with the role of the clock in promoting MuSC function, Bmal1 function is required for maintaining the proliferative and differentiative properties of myogenic progenitors under the chronic dystrophic disease milieu, which protects against the severity of dystrophic injury and functional decline to prolong survival [56]. In a dystrophic disease background, *Bmal1* deficiency resulted in markedly elevated serum creatine kinase levels across the age groups examined, indicative of global muscle damage. This is due to, at least in part, impaired regenerative capacity, as indicated by significant reduction in nascent myofiber formation. On the other hand, the loss of clock repressor Rev-erbα in the mdx background enhanced the regenerative myogenic response leading to protection against dystrophic muscle damage [55]. Notably, these results obtained from genetic models are consistent with the observed beneficial effects of Rev-erbα antagonist SR8278 in protecting against dystrophic damage in the mdx disease model [29]. These studies shed light on potential direct mechanistic links between the circadian clock and dystrophic disease, establishing a basis to test clock intervention strategies for muscular dystrophy treatment.

## 5. Circadian Clock as a Potential Drug Target for Duchenne Muscular Dystrophy

Dystrophic muscle in DMD patients present a range of pathophysiological features such as localized necrosis, inflammation, myofiber compensatory hypertrophy, deposits of connective tissue fibrosis, and fatty infiltrations [88]. While in normal muscles, satellite cell activation and regenerative repair maintain muscle homeostasis after injury, the chronic injury and inflammatory milieu in dystrophic muscle eventually leads to loss of muscle mass and function, partly due to insufficient regenerative capability to compensate for persistent muscle damage [54]. Clinically, DMD patients present with early-onset muscle wasting and weakness, with progressively deteriorating mobility. Worsening of respiratory failure needing ventilation leads to premature death between 20 and 40 years of age in these patients [118,119]. Current clinical therapy is largely limited to glucocorticoids to control chronic inflammation, with limited drug options available to patients despite increasing efforts to explore many experimental therapies currently in development [120]. There is an urgent need to develop new drugs for DMD, particularly as a combination therapy with current first-line corticosteroids.

Using mdx mice is an established pre-clinical mouse model for studying DMD [103,121,122,123]. They possess a point mutation in exon 23 of the *dystrophin* gene, producing a premature stop codon and resulting in the absence of full-length dystrophin protein [124]. Even though this model is mild compared to the DMD symptoms in humans, it recapitulates the shortened life span, high plasma levels of creatine kinase, necrosis, and muscle degeneration [125]. The mild dystrophic pathology in this model is likely due to a high regenerative capacity of the mdx mouse muscles, as compared to limited regenerative replication of the satellite cells of DMD patients. Additional models to better recapitulate the more severe human DMD dystrophic pathologies have been developed, including the dystrophin and utrophin double knockout model [126] or mdx mice lacking telomerase activity [127]. These genetic mutant models provide a closer resemblance to the phenotypic characteristics and disease progression of DMD, including loss of muscle force, increased serum creatine kinase levels, muscle fibrosis, and calcium deposits with a significant decline in MuSC regenerative capacity [127].

### 5.1. Current Research and Therapies for DMD

A strong focus of current research for muscular dystrophies is to better understand how sarcolemmal damage is initiated and repaired, with the ultimate goal of devising strategies to protect against the inciting injury through targeted interventions [89]. Gene therapy approaches to replace missing or defective dystrophin have been a major area of development. Due to size of the *dystrophin* gene, which spans nearly 2.3 megabases of the X chromosome, gene therapy options are limited in covering its entire coding sequence [128,129]. Significant research efforts are focused on alternative strategies [64]. Micro-dystrophin constructs were tested as a promising functional dystrophin therapy [130,131,132] with positive results from mdx mouse testing [133,134]. *Dystrophin*-based therapies also face significant challenges for the delivery of this large gene, and connective tissues surrounding the muscle prevent ready access to large viruses or macromolecules [135,136]. Cell therapies aiming at the transplantation of satellite cells with functional dystrophin have been explored with great potential for muscle replacement, while this is hindered by the failure of the majority of myoblasts to migrate into damaged muscle in situ [137,138]. Recent development of CRISPR-/Cas-based correction of DMD currently holds the strongest promise, with partial recovery of dystrophin functionality in mdx mice [139,140,141,142]. Non-dystrophin-based therapies such as upregulation of the *dystrophin* paralog, *utrophin*, also shows potential in mdx mice, although the dosage of utrophin required by patients to achieve a clinical benefit remains to be determined [143].

The current standard of care for DMD patients relies on glucocorticoids to maintain ambulation and improve muscle strength, but long-term treatments in young patients present severe side effects [108,120,144]. Among the adverse effects of glucocorticoids is the strong risk for muscular atrophies due to glucocorticoid stimulation of ubiquitin–proteasome systems, leading to proteolysis, an unwanted complication to avoid when treating muscular dystrophies [145,146]. Discovery of new drug therapies beyond current standard glucocorticoids for DMD is therefore keenly needed. Since the initial discovery of myostatin as a potent inhibitor of muscle growth [147,148], intense effort has been invested to block myostatin actions to treat muscle-wasting diseases, with therapeutic promise in ameliorating muscular dystrophies through pharmacological inhibition [149]. However, despite initial compelling results of myostatin inhibition in pre-clinical models [150,151], these efforts have largely been unsuccessful in clinical trials for DMD patients [152]. Recent studies offered mechanistic insights into the multifaceted role of circadian clock involvement in maintaining muscle mass, function, structural integrity, and satellite regenerative myogenic repair [18,25]. Given that these clock-controlled mechanisms are applicable for muscular dystrophy, therapeutic targeting of circadian clock-controlled pathways may mitigate the dystrophic phenotype that could hold promise for potential drug development efforts.

### 5.2. Recent Progress in Discovering Circadian Clock Modulators with Potential Applications for DMD

Accumulating research has firmly established that the circadian clock could be a novel therapeutic target for muscular dystrophy [27,54,56,58,153]. The molecular mechanisms underlying clock modulation of muscle mass, function, growth, and repair provide the mechanistic basis for exploring pharmacological interventions of clock activity to ameliorate dystrophic disease. Yet to date, our understanding of the complete suite of mechanistic pathways mediating the protective effects of the clock in dystrophic disease and its direct targets remains limited. New cistromic analysis revealed that components of the Igf-1 and integrin-associated signaling pathways are direct transcriptional targets of the molecular clock network [28,154]. Given that distinct clock transcriptional outputs are known DMD therapeutic targets [101,154], testing potential synergistic actions of clock-targeting interventions to promote muscle mass, function, regenerative repair, and sarcolemma stability may present unique opportunities for DMD drug development. Mechanistic insights garnered from genetic loss-of-function models suggest that reinforcement or activation of clock function may mitigate dystrophic pathophysiology [24,25]. Future exploration to modulate specific clock components amenable for dystrophic therapy is warranted [56], and ample opportunities remain to identify clock-targeting small molecules for DMD intervention.

Recent investigations have seen rapid progress toward developing clock-modulatory compounds for disease applications, particularly for cancer therapy [4]. A myriad of circadian clock modulators has been identified through the years that target Bmal1, CLOCK, or negative clock regulators, Crys, Pers, Rev-erbs, and RORs [10,155]. These agents constitute a list of potential clock modulators for the pharmacological intervention of the clock circuit for muscular dystrophies or related muscle diseases, as summarized in Table 1. Among them, Cry stabilizers such as KL001 [156], Rev-erb agonists such as SR9011 and SR9009 [157], or Rev-erbs antagonist SR8278 [158] have become new chemical probes for clock modulation with potential for drug development. Other molecules, currently in use for sleep disorders or dietary supplements such as melatonin [159,160,161] and nobiletin [162], could be further investigated for potential muscle disease indications.

Through high-throughput molecular docking followed by biochemical validations, chlorhexidine (CHX) was identified as a novel clock activator that displays strong pro-myogenic activities [11]. By activating Bmal1-/CLOCK-mediated transcription, CHX induces myogenesis by promoting Wnt signaling activity, a clock-controlled output pathway in myoblasts. CHX treatment of myoblasts was able to enhance distinct stages of myocyte development, including proliferation, differentiation, and migration [11]. This study demonstrated the feasibility and utility of a screening pipeline to discover new molecules with clock-modulatory properties to promote myogenesis, by directly targeting the CLOCK protein for the first time. Given the activities of CHX in promoting clock-dependent myogenesis, observed in myoblasts, it is conceivable that a similar action in vivo may augment regenerative capacities. Future in vivo studies of CHX or its derivatives with improved efficacy are warranted to explore their therapeutic potential for dystrophic or degenerative muscle diseases. As a first step, testing CHX using the mdx model may establish its applicability for dystrophic disease. Due to the chronic inflammatory milieu in dystrophic conditions, it is possible that additional mechanisms of clock modulation may apply for DMD, such as known clock regulatory effects on macrophage-mediated inflammation or immune regulations [173,174,175]. Particularly, chronic inflammation in dystrophic diseases is in large part due to macrophage infiltration and thus a significant contributor to the severity of DMD pathology [176], while circadian clock is intimately involved in innate immune regulation [175]. In addition, RORs and Rev-erbα/β play key roles in modulating T lymphocyte function [177,178], and synthetic ligands for these clock regulators are available and currently being explored for applications in inflammatory disease conditions [169,179,180]. Hence, small molecules targeting circadian clock regulators that may possess immune-modulatory properties in addition to pro-myogenic actions could be developed to promote regenerative capacity while alleviating chronic inflammation in DMD, which may yield robust efficacy in ameliorating muscular dystrophy. Also conceivable is that structural optimization of the CHX chemical scaffold may yield additional molecules with improved clock-activating properties. Medicinal chemistry efforts to obtain new analogs could yield compounds with improved pro-myogenic efficacy while minimizing toxicity toward the next stage of muscular dystrophy drug development. Clock-targeting compounds could also be applicable to diseases involving defective muscle remodeling, such as aging-associated sarcopenia. Drug development efforts are needed for the therapeutic targeting of the clock circuitry as alternative interventional strategies to benefit DMD patients. Furthermore, recent advances in bioengineering approaches, such as artificial muscle constructs or organs-on-a-chip that incorporate cell–cell communications via microfluidic technology to mimic muscle tissue ex vivo, could be applied for drug discovery efforts for clock modulators or leveraged for DMD disease modelling [181]. The advent of these emerging technologies could be adopted to develop clock-targeted therapies, for example, testing CHX and its derivative in a physiologically relevant disease model for DMD.

## 6. Future Perspectives

Continued research to gain a comprehensive mechanistic understanding of the role of the circadian clock in muscle disease development and progression will provide the basis for clock-targeted therapies. Targeting the circadian clock for dystrophic disease drug development may yield novel agents for combinatorial therapy with glucocorticoids. Given the distinct roles of clock components in muscle biology and disease involvement, therapeutic targeting of specific clock regulators may offer new opportunities for much-desired pharmacological options for DMD patients (Figure 5). Future availability of clock-targeted intervention strategies could translate into clinical impacts on disease severity and ultimately mitigate the devastating outcomes of DMD.

## Figures and Tables

**Figure 1 ijms-25-04767-f001:**
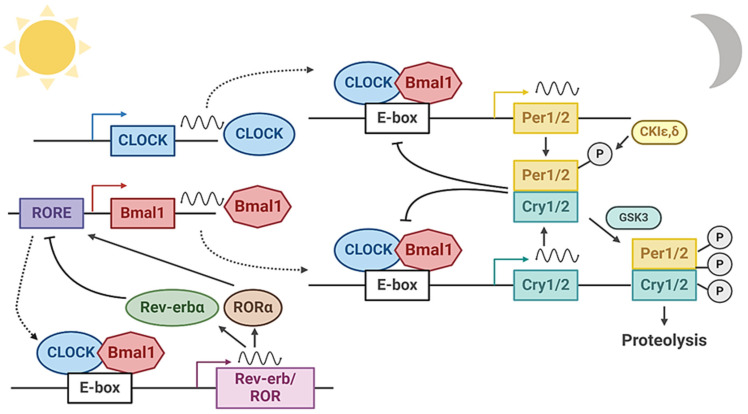
Components of the Transcriptional–Translational Feedback Loop (TTFL) of the molecular clock. The core circadian clock machinery is composed of interlocking feedback loops of transcriptional and translational regulation, with transcription activation by CLOCK and Bmal1 that is inhibited through Cry- and Per-mediated repression. Post-translational regulatory mechanisms of Per and Cry are involved in the de-repression of the core clock TTFL. Rev-erbα- and RORα-mediated transcriptional control of Bmal1 rhythmic expression constitutes an additional stabilizing loop within the clock’s molecular network. Clock proteins are represented in colored boxes. Arrows denote activation, and blocked lines denote repression.

**Figure 2 ijms-25-04767-f002:**
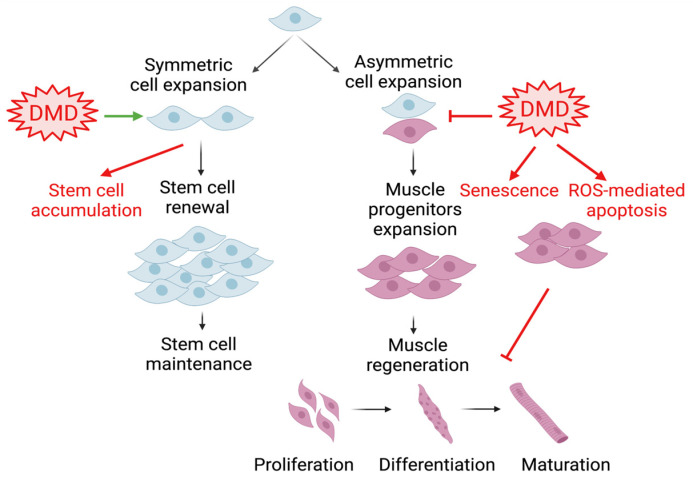
Muscle stem cell dysfunction in DMD. Satellite cell expansion through symmetric and asymmetric division is critical to maintaining a muscle stem cell pool while providing a committed progenitor population for muscle regeneration. The imbalance of these processes in DMD leads to stem cell accumulation with insufficient progenitors. In addition, other aspects of stem cell dysfunction, including senescence and mitochondrial reactive oxygen species, collectively contribute to the impaired regenerative capacity in DMD. Arrows denote activation, and blocked lines denote repression.

**Figure 3 ijms-25-04767-f003:**
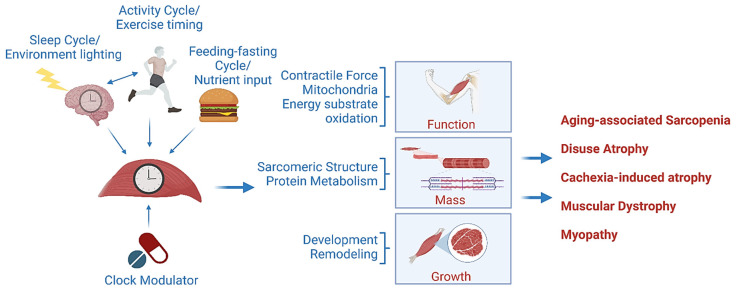
Clock modulation of muscle mass, function, and growth with disease implications. The circadian clock confers temporal control in muscle metabolism, growth, and remodeling. Muscles intrinsic clock receives entrainment cues from the central clock, diurnal activity cycles, and nutrient oscillations derived from feeding and fasting. Clock output pathways determine time-of-day variations in muscle function, maintain muscle mass, and modulate growth processes. Disruption of these mechanisms may contribute to a myriad of muscle disease conditions as a result of clock dysregulation, including muscle atrophy, aging-associated sarcopenia, or muscular dystrophy. The muscle clock is thus amenable to pharmacological interventions for disease therapy.

**Figure 4 ijms-25-04767-f004:**
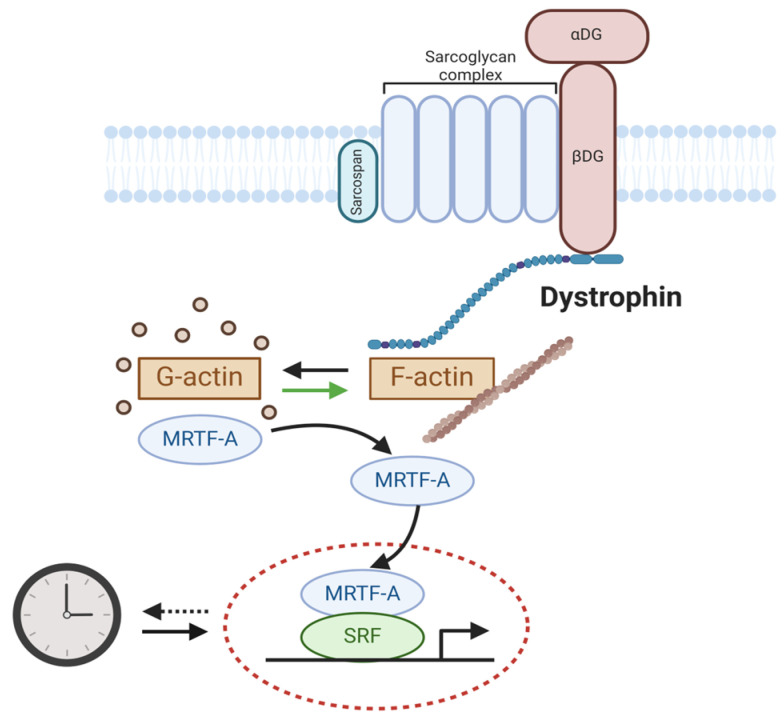
Mechanisms linking the circadian clock with dystrophin intracellular signaling. Dystrophin connects the intracellular actin cytoskeleton with the sarcolemma and extracellular matrix. Actin cytoskeleton dynamics induce MRTF-A nuclear translocation, leading to SRF-mediated transcriptional activation of core clock genes. The circadian clock exerts reciprocal transcriptional control of key components of intracellular actin remodeling and MRTF-A/SRF signaling. Arrows denote activation.

**Figure 5 ijms-25-04767-f005:**
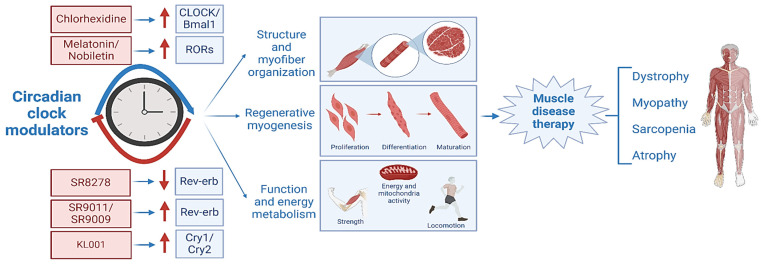
Small-molecule clock modulators and their therapeutic potential for muscle disease applications. Specific components of the circadian clock circuit in skeletal muscle can be targeted by clock-modulatory small molecules. Their mechanisms of action involve disparate clock-controlled processes in maintaining muscle structure, myofiber organization, myocyte development, or muscle energy metabolism. These clock modulators may have therapeutic applications in various muscle disease conditions, such as muscular dystrophy or age-associated sarcopenia. 
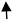
: denotes activation and 
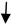
 denotes inhibition.

**Table 1 ijms-25-04767-t001:** Effects of clock-modulatory compounds on muscle function and disease implication.

Compound	Clock Target	Clock-Modulatory and Associated Effects	Muscle Disease Relevance	Human Studies	References
SR8278	Rev-erbα/β	Antagonist Activation of clock gene expression	Stimulation of muscle regeneration Promotion of myoblast proliferation and differentiation	N/A	Bugge et al. 2012 [163], Chatterjee et al. 2019 [53], Kiperman et al. 2023 [11], Kojetin et al. 2011 [158], Welch et al. 2017 [59]
SR9011/SR9009	Rev-erbα/β	Agonist Suppression of clock genes Induction of wakefulness Increase in energy expenditure and weight loss	Increase in mitochondrial content Promotion of muscle oxidation	N/A	Amador et al. 2016 [164], Fan et al. 2013 [165], Geldof et al. 2016 [166], Solt et al. 2012 [157], Woldt et al. 2013 [42]
Chlorhexidine	CLOCK	Activator Promotion of Bmal1/CLOCK interaction Induction of clock gene expression	Promotion of satellite cell proliferation and myogenic differentiation	Antiseptic/antimicrobial, mouth wash	Chatterjee et al. 2013 and 2015 [24,25], Kiperman et al. 2023 [11]
N-acetyl-5-methoxytryptamine (Melatonin)	RORα/γ	Inhibition of ubiquitin–proteasome protein degradation to modulate circadian clock loop Stabilization of circadian rhythm	Mitigation of age-related sarcopenia	Insomnia, sleep-related pathologies, cancer, neuroprotection	Becker-André et al. 1994 [159], Fernández-Martínez et al. 2023 [160], Jetten 2009 [167], Sayed et al. 2018 [168]
SR1078	RORα/γ	Agonist Promotion of core clock target gene expression	Not determined	N/A	Kojetin et al. 2010 [158]
SR1001	RORα/γ	Inverse agonist Inhibition of clock oscillation	Not determined	N/A	Solt et al. 2011 [169]
SR3335	RORα	Selective inverse agonist Suppression of RORα target gene expression Inhibition of gluconeogenesis	Not determined	N/A	Kumar et al. 2012 [170]
Nobiletin	RORα/γ	Agonist Enhancement of clock oscillatory amplitude Activation of circadian clock Promotion of energy homeostasis	Increase in muscle mitochondrial oxidation Promotion of lipid metabolism	Weight loss, Alzheimer’s disease	He et al. 2016 [162], Raichur et al. 2010 [171]
KL001	Cry1/2	Cry1/2 stabilizer Period lengthening Inhibition of glucagon-induced gluconeogenesis	Not determined	Hemophilia B	Hirota et al. 2012 [156], Nangle et al. 2013 [172]

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
