# Peer review of "Circadian Clock in Muscle Disease Etiology and Therapeutic Potential for Duchenne Muscular Dystrophy"

_ijms, 2024, doi:10.3390/ijms25094767_

Round 1
Reviewer 1 Report
Comments and Suggestions for Authors
This is an interesting and well-written article in which the authors reviewed the importance of the circadian clock in muscle physiology, as well as its relevance in pathological muscle conditions such as Duchenne Muscular Dystrophy and potential therapeutic approaches.
In my opinion the article deserves publication, I just have some minor comments.
Please revise and be consistent with correct nomenclatures for human protein/gene abbreviations:
https://www.ncbi.nlm.nih.gov/genome/doc/internatprot_nomenguide/
https://www.ncbi.nlm.nih.gov/pmc/articles/PMC7494048/
Some efforts to block myostatin function to ameliorate musclular dystrophy have been made. Do myostatin inhibitors or anti-myostatin recombinant proteins show promise for the treatment of DMD? Is there any evidence of interaction between clock proteins and myostatin?
Animal models for muscular dystrophy have provided important information to understand the pathology; however there are limitations to drug discovery and therapy due to species-specific differences between model animals and humans. An organ-on-a-chip could be a more effective approach to gain insights into drug discovery and therapies for DMD (e.g. to test CHX), please discuss whether this could be an option for developing clock-targeted therapies.
Author Response
We sincerely appreciate the reviewer’s careful review of our manuscript. We are gratified that the reviewers found that the current manuscript “provides a comprehensive account of clock functions within skeletal muscle. We also appreciate their insightful suggestions to improve this review, and accordingly, we revised the manuscript based on the specific comments. A point-to-point response to the questions raised is provided below.
Reviewer 1:
This is an interesting and well-written article in which the authors reviewed the importance of the circadian clock in muscle physiology, as well as its relevance in pathological muscle conditions such as Duchenne Muscular Dystrophy and potential therapeutic approaches.
- Please revise and be consistent with correct nomenclatures for human protein/gene abbreviations:
Response: We have revised the specific gene and protein names according to the guidelines.
- Some efforts to block myostatin function to ameliorate muscular dystrophy have been made. Do myostatin inhibitors or anti-myostatin recombinant proteins show promise for the treatment of DMD? Is there any evidence of interaction between clock proteins and myostatin?
Response: The reviewer raised a very relevant question. Accordingly, we have revised our manuscript to include discussion of myostatin inhibitors/monoclonal antibodies for the treatment of DMD. Since the initial discovery of myostatin as a potent inhibitor of muscle growth, intensive effort has been invested to block myostatin actions to treat muscle wasting diseases. However, despite initial compelling results in pre-clinical models, these efforts have largely been unsuccessful in clinical trials for DMD patients. This discussion is included in the revision with added reference (Page11, line 458-463).
To date, there has not been reported interactions between clock protein and myostatin. Interestingly, beta2-AR agonist formoterol treatment attenuated expression of genes involved in myostatin signaling, with concomitant altered clock gene regulation including DBP, Nfil3 and Cry2 (PMID: 19772666).
- Animal models for muscular dystrophy have provided important information to understand the pathology; however there are limitations to drug discovery and therapy due to species-specific differences between model animals and humans. An organ-on-a-chip could be a more effective approach to gain insights into drug discovery and therapies for DMD (e.g. to test CHX), please discuss whether this could be an option for developing clock-targeted therapies.
Response: We appreciate the reviewer’s insightful suggestion. We also agree that by incorporating cell-cell communications via microfluidic technology using distinct cell types found within a specific tissue, the recent advent of organ-on-the-chip offers a novel modality for drug discovery. This technology could be potentially leveraged for DMD disease modelling to include extracellular matrix, vasculature, immune cells in addition to myotubes to better recapitulate the complex in vivo DMD pathophysiology. This bio-engineering approach could be used to develop clock-targeted therapies, for example to test the possibility of CHX or related compounds in a physiologically relevant disease model for DMD. We added these discussions in Page 13 line 535-541.
Reviewer 2 Report
Comments and Suggestions for Authors
The article examines the role of circadian rhythms in the pathogenesis of a specific genetic muscular atrophy known as Duchene Muscular Dystrophy (DMD).
Specific comments are listed below.
1. Introduction - Usually lays out the general background and aim of the review. However, since the review focuses on DMD, a few sentences linking DMD (prevalence, pathogenesis, etc.) to regulation of the circadian clock will be helpful.
2. Circadian clock regulation of skeletal muscle structure and function - the information is sufficient and precise. Comments – is figure-1 original or adapted? If modified from a previous publication, the original article source needs to be cited.
3. Circadian clock in muscle stem cell biology – a) relevant evidence of aberrant muscle stem cell pathology in DMD, b) a figure that captures the essence of this discussion would be helpful.
4. Circadian clock dysregulation in DMD – Since this review focuses on DMD, you need to elaborate more on the role of dystrophin involvement in peripheral circadian SRF. Most of the information has been presented (Bettes CA, et al., Life Sci Alliance, 2021 and Rossi R, et al. Frontiers in Physiology, 2021), you just need to re-organize the information. A figure summarizing the information would also be helpful.
Comments on the Quality of English LanguageNo further comment for this aspect. Language proficiency is sufficient.
Author Response
We sincerely appreciate the three reviewer’s careful review of our manuscript. We are gratified that the reviewers found that the current manuscript “provides a comprehensive account of clock functions within skeletal muscle. We also appreciate their insightful suggestions to improve this review, and accordingly, we revised the manuscript based on the specific comments. A point-by-point response to the questions raised is provided below.
Reviewer 2:
- Introduction - Usually lays out the general background and aim of the review. However, since the review focuses on DMD, a few sentences linking DMD (prevalence, pathogenesis, etc.) to regulation of the circadian clock will be helpful.
Response: We appreciate the reviewer’s thoughtful comment and added this component in the Introduction section as suggested (Page 1 Line 34 -39).
- Circadian clock regulation of skeletal muscle structure and function - the information is sufficient and precise. Comments – is figure-1 original or adapted? If modified from a previous publication, the original article source needs to be cited.
Response: Figure 1 is original.
- Circadian clock in muscle stem cell biology – a) relevant evidence of aberrant muscle stem cell pathology in DMD, b) a figure that captures the essence of this discussion would be helpful.
Response: a) As reviewer suggested, we added discussion of aberrant muscle stem cell pathology in DMD (Page 4 line 134-142), and included additional key references (Ref#50-52). b) A new Figure 2 is added to illustrate relevant evidence of aberrant muscle stem cell pathology in DMD as reviewer suggested.
- Circadian clock dysregulation in DMD – Since this review focuses on DMD, you need to elaborate more on the role of dystrophin involvement in peripheral circadian SRF. Most of the information has been presented (Bettes CA, et al., Life Sci Alliance, 2021 and Rossi R, et al. Frontiers in Physiology, 2021), you just need to re-organize the information. A figure summarizing the information would also be helpful.
Response: We appreciate the reviewer’s suggestion and revised this section accordingly. A new Figure 3 is added to further illustrate the discussion.
Reviewer 3 Report
Comments and Suggestions for Authors
The paper authored by Kiperman and colleagues provides a comprehensive account of clock functions within skeletal muscle. Notably, the intricate relationship between the synthesis and degradation of skeletal muscle proteins, satellite cell functions, and the circadian clock is meticulously elucidated. I recommend further elaboration on certain aspects to aid the reader's comprehension.
1. Could the authors elaborate on why the circadian clock system experiences disruption upon the onset of DMD? Are muscle contractions or physical stimuli responsible for entraining the circadian clock? Additionally, is the diminished circadian clock function observed in DMD a causal factor in dystrophy onset, or does it manifest as a consequence of atrophy?
2. The principal trigger of muscular dystrophy is the necrosis of skeletal muscle, significantly influenced by the immune system. It would greatly benefit readers if the authors could expound upon the effects of clock function disruption on the immune system and its correlation with muscular dystrophy, as indicated in Lines 478-481.
3. The authors are providing methods to ameliorate muscular dystrophy by rectifying clock function, showing that Compound acts on several clock molecules (Table 1). I am curious to know the extent of progress made in human studies concerning the therapeutic application of these molecules, not limited to muscular dystrophy.
Author Response
We sincerely appreciate the three reviewer’s careful review of our manuscript. We are gratified that the reviewers found that the current manuscript “provides a comprehensive account of clock functions within skeletal muscle. We also appreciate their insightful suggestions to improve this review, and accordingly, we revised the manuscript based on the specific comments. A point-by-point response to the questions raised is provided below.
Reviewer 3:
The paper authored by Kiperman and colleagues provides a comprehensive account of clock functions within skeletal muscle. Notably, the intricate relationship between the synthesis and degradation of skeletal muscle proteins, satellite cell functions, and the circadian clock is meticulously elucidated. I recommend further elaboration on certain aspects to aid the reader's comprehension.
- Could the authors elaborate on why the circadian clock system experiences disruption upon the onset of DMD? Are muscle contractions or physical stimuli responsible for entraining the circadian clock? Additionally, is the diminished circadian clock function observed in DMD a causal factor in dystrophy onset, or does it manifest as a consequence of atrophy.
Response: We appreciate the reviewer’s insight. Betts et al. (Ref #102) revealed that the loss of dystrophin and associated connection with intracellular actin cytoskeleton in DMD altered MRTF-A/SRF signaling, which is a known pathway in entraining circadian clock. Previous study from our group has shown that extracellular physical niche cues, such as matrix stiffness, can be transduced intracellularly via the actin cytoskeleton and MRTF-A/SRF signaling to entrain the circadian clock. In addition, exercise is known to lead to phase shift muscle clock, suggesting potential muscle contraction and mechanical force involved in modulating clock function. It remains to be determined whether altered clock function in DMD could play a causal role in dystrophy pathogenesis, or occurred as a consequence of dystrophic changes in muscle. We have now added this discussion in the revision (Page 8 Line 331-347).
- The principal trigger of muscular dystrophy is the necrosis of skeletal muscle, significantly influenced by the immune system. It would greatly benefit readers if the authors could expound upon the effects of clock function disruption on the immune system and its correlation with muscular dystrophy, as indicated in Lines 478-481.
Response: As reviewer suggested, we further expanded the discussion of the effects of clock function disruption on the immune system as related to muscular dystrophy in page 12 line 518-527.
- The authors are providing methods to ameliorate muscular dystrophy by rectifying clock function, showing that Compound acts on several clock molecules (Table 1). I am curious to know the extent of progress made in human studies concerning the therapeutic application of these molecules, not limited to muscular dystrophy.
Response: Human studies concerning the therapeutic applications of these clock-modulatory molecules remain limited so far. Nonetheless, as reviewer suggested, we performed a search for clinical studies and added the relevant information regarding in an additional column in Table 1.